# LC-HRMS Screening and Identification of Novel Peptide Markers of Ricin Based on Multiple Protease Digestion Strategies

**DOI:** 10.3390/toxins11070393

**Published:** 2019-07-05

**Authors:** Long-Hui Liang, Chang-Cai Liu, Bo Chen, Long Yan, Hui-Lan Yu, Yang Yang, Ji-Na Wu, Xiao-Sen Li, Shi-Lei Liu

**Affiliations:** 1State Key Laboratory of NBC Protection for Civilian, Beijing 102205, China; 2The laboratory of Analytical Chemistry, Research Institute of Chemical Defence, Beijing 102205, China

**Keywords:** ricin, marker peptides, unambiguous identification, mass spectrometry

## Abstract

Both ricin and *R. communis*
*agglutinin* (RCA120), belonging to the type II ribosome-inactivating proteins (RIPs-Ⅱ), are derived from the seeds of the castor bean plant. They share very similar amino acid sequences, but ricin is much more toxic than RCA120. It is urgently necessary to distinguish ricin and RCA120 in response to public safety. Currently, mass spectrometric assays are well established for unambiguous identification of ricin by accurate analysis of differentiated amino acid residues after trypsin digestion. However, diagnostic peptides are relatively limited for unambiguous identification of trace ricin, especially in complex matrices. Here, we demonstrate a digestion strategy of multiple proteinases to produce novel peptide markers for unambiguous identification of ricin. Liquid chromatography-high resolution MS (LC-HRMS) was used to verify the resulting peptides, among which only the peptides with uniqueness and good MS response were selected as peptide markers. Seven novel peptide markers were obtained from tandem digestion of trypsin and endoproteinase Glu-C in PBS buffer. From the chymotrypsin digestion under reduction and non-reduction conditions, eight and seven novel peptides were selected respectively. Using pepsin under pH 1~2 and proteinase K digestion, six and five peptides were selected as novel peptide markers. In conclusion, the obtained novel peptides from the established digestion methods can be recommended for the unambiguous identification of ricin during the investigation of illegal use of the toxin.

## 1. Introduction

Type Ⅱ ribosome-inactivating proteins (RIPs-Ⅱ) are a class of heterodimeric protein toxins, which consists of two disulfide-linked polypeptide chains. The A-chain is a functional chain of N-glycosidase activity responsible for irreversibly inactivating the large ribosome subunits through depurination of a specific adenine from 28S rRNA. The B-chain has lectin activity on binding to the terminal galactose receptors on cell surface allowing for A-chain internalization by endocytosis [1,2]. Ricin is one of the best known RIP-Ⅱtoxins, which was extracted from the castor bean plant *Ricinus communis*. Exposure symptoms include nausea, vomiting, fever and even death [3]. Due to its wide availability, high toxicity and ease of preparation, ricin has been used in several threat incidents since the beginning of the 20th century [4]. As one of the most potent chemical warfare agents and bioterrorism agents, production and possession of ricin have been prohibited by the Chemical Weapons Convention (CWC) since 1997 [5]. In response to the potential use of ricin for bioterrorism, criminal, or military purposes, establishing accurate methods is very necessary for the identification of ricin and other RIP-Ⅱ protein toxins.

Immunological and functional assay methods have been developed to determine ricin and other RIP-II toxins [6,7,8,9,10]. However, the presence of *R. communis agglutinin* (RCA120) makes it difficult for these methods to unambiguously identify ricin due to its identical functional activity and high amino acid sequence similarity (93% in A-chain and 84% in B-chain) with ricin. Furthermore, because of the lower toxicity of RCA120 than ricin (2~3 orders) [4,11], it is very essential to distinguish ricin from RCA120 for the unambiguous identification of ricin.

Mass spectrometric (MS) techniques provide an alternative approach for unambiguous identification of ricin by accurate determination of amino acid sequence differences between ricin and RCA120. Before MS detection, it is essential to produce unique peptide markers of ricin with suitable molecular weight and enough MS response using proteinase digestion. Currently, tryptic peptides of T7A, T11A, T6B and T18B unique to ricin have become the common and representative peptide markers [12,13,14,15]. The peptide markers were widely used for unambiguous identification of ricin in complex matrices including crude extracts of castor bean, protein and stevia powder, beverages and soil, et al. [16]. Some efforts were made to enhance the efficiency of trypsin digestion. For example, instead of trypsin digestion following the denaturation, reduction and alkylation, the direct trypsin digestion of intact ricin under the solvent-assisted condition was performed with no missed cleavage and reduced digestion time [12,17]. Besides, two peptide markers (C18A and C18B) generated from chymotrypsin digestion were also applied for unambiguous identification of ricin from crude extracts of castor bean and soil samples [18]. However, the selected peptide markers are still relatively limited. Thus, it is necessary to obtain abundant peptide markers produced from more types of proteinase digestion for unambiguous identification of trace ricin, especially in complex matrices.

Here, novel peptide markers of ricin were screened and identified using LC high-resolution (HR) MS based on tandem and multiple protease digestion strategies. Typical proteinases including trypsin, Glu-C, chymotrypsin, pepsin, and proteinase K were attempted to generate abundant peptides. The digested peptides from different proteases were screened using In-Silico BLASTp search against a non-redundant protein database. The unique peptides with a positive MS response were screened and identified by LC-HRMS and selected as peptide markers. 

## 2. Results and Discussion

### 2.1. Amino Acid Sequence Difference between Ricin and RCA120

To understand the sequence differences between ricin and RCA120, sequence alignment is performed using the Clustal W program of BioEdit 7.0.9.0 (Figure 1). The two ricin isoforms of D and E have an identical amino acid sequence in the A-chain, only 15% sequence difference in the B-chain. Ricin E is considered as one gene recombination product between ricin D and RCA120, because of its identical N-terminus in B-chain with that of ricin D and the identical C-terminus in the B-chain with that of RCA120 [19]. Based on their identical sequence of A-chain, ricin D and E share similar toxicity. Meanwhile, because of the extremely low content of ricin E in all *R. communis* ecotypes, “ricin” is specially referred to as ricin D throughout this article. 

RCA120 shows high sequence homology with ricin D, with 93% for the A-chain and 84% for the B-Chain (Figure 1). It has the same N-glycosidase activity of A-chain and lectin activity of B-chain with ricin, but lower toxicity. Therefore, it is of great importance to discriminate ricin and RCA120 in complex matrices such as castor bean extracts. Currently, MS-based technique can be used to detect and distinguish between ricin and RCA120 through the small number of different amino acid sequences in peptide markers.

### 2.2. Ricin Peptide Markers from Trypsin Digestion

Peptide markers of ricin from trypsin digestion have been mainly produced by digestion after denaturation, reduction and alkylation [13,20], or direct digestion of intact ricin [12,17]. To fully understand the profile of tryptic peptide markers of ricin, efforts were made to screen them by digestion under both reduction and non-reduction. Tryptic peptides unique to ricin D (P02879) were screened by BLASTp searching tryptic peptides across all protein sequences from the updated UniprotKB/Swiss-Prot database. In silico digestion of ricin was carried out using the peptide mass tool (https://web.expasy.org/peptide_mass/) to obtain tryptic peptides. Tryptic peptides unique to ricin across all protein sequences were considered as candidates of its peptide markers. There are nine specific peptides identified from both the A-chain and B-chain under the reduction condition (Appendix A); Nine specific peptides in A-chain and eight specific peptides in B-chain were identified under the non-reduction condition (Appendix A). Under both conditions, TA5 (LTTGADVR) and TB19 (ASDPSLK) had been used as peptide markers for differentiating ricin from RCA120 [12,20], but they were confirmed not to be unique since they are also present in the proteins from apple snail (P83672) and winter moth (A0AL7KVU4). Taking account of this reason, the two peptides of TA5 and TB19 were not considered as candidates of ricin peptide markers.

Subsequently, tryptic digestion of purified ricin D standard was performed using our improved method of tryptic digestion under the condition of denaturation and reduction, in which one denaturant resistant trypsin/lys-C mix (Promega) was applied to reduce enzymatic activity inhibition and missed cleavage. To reduce the labor procedure and elevate ionization efficiency, acetonitrile (Acn) of 20% was used as a denaturation agent instead of urea or guanidine chloride. Besides, the traditional alkylation step using iodoacetamide (IAM) modification was removed because of the low percentage of cysteine residues. As a result, all ricin specific peptides were detected by LC-quadrupole-time of flight-MS (LC-Q/TOF-MS) except for T12B, probably due to its longest length of 67 amino acid residues. The most sensitive digested peptides in order were T7A, T11A, T9A, T10A, T13A and T12A in A-chain, T20B, T6B, T14B and T18B in B-chain. These unique peptides were considered as the peptide markers suitable for unambiguous identification of ricin. The three tryptic peptides of T10A, T6B and T18B had not been selected as peptide markers of ricin [20], probably due to missed cleavage by non-denaturant resistant trypsin. The use of the trypsin/lys-C mix resistant to denaturant reduced the missed cleavage and increase the yields of these peptides. Direct tryptic digestion of purified ricin D standard was performed using our unpublished 10% Acn-assisted trypsin digestion system. It was found that similar peptide markers were obtained with the top four MS abundant peptides of T9A, T7A, T6B and T20B. The unique intra-disulfide linked peptides of both T3B-ss-T5B and T14B-ss-T16B were identified. In addition, one common inter-disulfide T24A-ss-T1B between intact ricin and RCA120 was also detected, which has been considered as an important peptide marker for the presence of intact ricin or RCA120 [17].

### 2.3. Novel Peptide Markers of Ricin from Tandem Digestion with Endoproteinase Glu-C after Trypsin

Followed by the above trypsin digestion, several specific peptides have been selected as peptide markers with good MS response. However, some unique peptides (e.g., T8A, T12A and T10B) with molecular weight (MW) over 3 kDa are not suitable as peptide markers due to their lower MS response. Notably, we found in these larger unique peptides there were common aspartic acid (D) or glutamic acid (E) residues, which of C-terminus can be specially cleaved by endoprotease Glu-C. To confirm this, firstly, the mixed proteinase system with both trypsin and Glu-C was attempted to digest these large peptides in NH_4_HCO_3_ buffer. However, the result was not satisfied probably due to the low cleavage efficiency of Glu-C in NH_4_HCO_3_ buffer. Tandem digestion strategy was then introduced to digest the large peptides by Glu-C in phosphate buffer after trypsin digestion in NH_4_HCO_3_ buffer according to 3.2.3 protocol. In silico Glu-C digestion followed by trypsin shows more unique peptides than separate tryptic digestion (Table 1, Appendix A). It was found that in silico digested peptides using Glu-C after trypsin were well consistent with the resulting peptides analyzed by LC-Q/TOF-MS by their extracted ion chromatographs (Figure 2).

As a result, we obtained three more novel peptide markers of TG13A, TG26A and TG28A in A-chain with tandem digestion strategy than with single trypsin digestion. The limits of detection (LODs) of ricin for TG13A, TG26A, TG28A were 50 ng/mL, 0.05 mg/mL and 50 ng/mL respectively. There is no D and E residue sites in the amino acid sequence of T2A-glyc and T7A, the two peptides after Glu-C digestion were respectively denominated as TG2A-glyc and TG9A. The larger unique peptides such as T8A and T12A were subsequently digested by Glu-C into shorter peptides suitable for MS analysis. The uniqueness of shorter peptides is re-confirmed by BLASTp search all other proteins. Amongst these unique peptides, TG13A from T8A, as well as TG26A and TG28A from T12A in A-chain were identified as novel peptide markers with good MS response. Because of the strong hydrophobic property, T13A from tryptic digestion has been confirmed to exhibits poor chromatographic behavior and MS response [12]. After subsequent Glu-C digestion, T13A was cleaved into TG29A with weaker hydrophobic property than T13A based on Kyte-Dolittle hydropathic index analysis. Correspondingly, TG29A exhibits better chromatographic behavior than T13A but the MS response is still relatively low. In addition, the TG20A from T10A had a relatively low MS response, probably due to the common intermediate resistance at the region of DR by proteinase cleavage [12].

In B-chain, four more novel peptide markers of TG6B-ss-TG9B, TG11B, TG27B-ss-TG30B, and TG37B were obtained, and their LODs were 100 ng/mL, 100 ng/mL, 0.05 mg/mL and 100 ng/mL respectively. In details, following Glu-C digestion, the two tryptic peptides of T18B and T20B were cleaved into the three peptides of TG33B, TG37B, and TG38B. The peptide of TG11B with seven amino acid residues from T6B was also confirmed to be unique and possess good response in MS analysis. We also identified two novel peptides of TG6B-ss-TG9B and TG27B-ss-TG30B with intramolecular disulfide bond in MS analysis. The peptide of T10B should have been transferred into a novel intramolecular disulfide bond peptide of TG15B-ss-TG16B, but it was not observed in MS analysis probably due to its low ionization efficiency. Notably, the AB chain linker peptide TG47A-ss-TG2B from T24A-ss-T1B was identified with comparably chromatography retention and MS response with TG26A. Similar to T24A-ss-T1B, TG47A-ss-TG2B also can provide an important information for the presence of an intermolecular disulfide bond in spite of the consistence between ricin and RCA120.

The applicability and selectivity were estimated by determination of one homologous mixed protein sample (0.25 mg/mL of ricin, RCA120 and abrin) and one sample of 0.5 mg/mL RCA120. It was found that all seven ricin marker peptides were clearly identified in the homologous mixed protein sample. As for 0.5 mg/mL RCA120, no background/interference peaks of ricin marker peptides were detected, suggesting good selectivity of this method allowing for discrimination of RCA120 from Ricin. 

### 2.4. Novel Peptide Markers of Ricin from Chymotrypsin Digestion

Recently, one LC-MS method combined with chymotrypsin digestion was developed for identification of ricin [18]. Two specific peptides of C18A (TDVQNRY) and VL-C18B (VL-AATSGNSGTTL) with one missed cleavage site were selected as peptide markers for accurate identification of ricin in plant extracts and soil samples [18]. However, the selected peptide markers are limited for unambiguous identification of ricin in a complicated matrix. In addition, we found that in our digestion system, the MS response for VL-C18B was low because of missed cleavage at the first Leucine residue using the previous method of chymotrypsin digestion. To completely screen chymotryptic peptide markers, chymotrypsin digestion of ricin under both reduction and non-reduction conditions were optimized and performed followed by LC-Q-TOF/MS analysis of the digested peptides. The uniqueness of each chymotryptic peptide was verified by BLASTp searching against the NCBI UniprotKB/Swiss-Prot protein database. As a result, we find that there were seven specific peptides of C2A, C4A, C5A, C18A, C27A, C31A and C33A in A-chain, and 14 specific peptides of C2B-C5B, C8B, C14B, C16B, C18B, C26B, C27B, C31B, C40B, C41B and C43B in B-chain under digestion with denaturation and reduction (Table 2). As for non-reduction conditions, besides the above specific peptides, both the AB chain linker peptide of C49A-ss-C1B (RCAPPPSSQF-ss-ADVCM) and the intramolecular disulfide bond peptide of C3B-ss-C5B (CVDVRDGRF-ss-WPCKSNTDANQL) were different peptides from digestion with denaturation and reduction. 

Under the condition of denaturation and reduction, peptide mapping analyses were performed using our optimized digestion conditions with 20% Acn as denaturation agent to cleave purified ricin. In A-chain, all 7 specific chymotryptic fragments were verified using LC-MS analysis as shown in Figure 3 and listed in Table 2. The three peptides of C18A, C4A and C31A with 7–10 amino acid residues were selected as novel peptide markers based on suitable MW and good MS response, whose LODs were 10 ng/mL, 20 ng/mL, and 50 ng/mL respectively. In the B-chain, 11 out of 14 specific chymotryptic peptides were identified in MS analysis (Figure 3) and listed in Table 2. The chymotryptic peptides with the best MS response were C8B, C27B, C31B, C18B, and C4B in order and their LODs were 20 ng/mL, 20 ng/mL, 50 ng/mL, 50 ng/mL and 100 ng/mL respectively. They were considered as the peptide markers suitable for unambiguous identification of ricin. Notably, we observed no missing cleavage peptides of C18B AATSGNSGTTL, suggesting that the better accuracy of digestion was achieved than that of previous studies. Both C40B and C43B peptides were not identified in MS probably because they were located in the polar region in B-chain by Kyte-Dolittle hydropathic index analysis. This result is consistent with previous findings that polar regions may be more resistant to digestion than other regions of protein [12].

Under the condition of Acn-assisted direct digestion, similar number of chymotryptic fragments were selected as peptide markers including three peptide markers of C18A, C4A and C31A in A chain, as well as four peptide markers of C18B, C8B, C41B and C31B in B chain (Figure 4). An exception for this condition was that the unique intra-disulfide linked peptide C3B-ss-C5B was identified in the extraction ion chromatography of the triply charged molecular ion at 814.0431. The common inter-chain disulfide peptide C49A-ss-C1B between ricin and RCA120 was also detected in extracted ion chromatograms (Figure 4). Although both the inter-chain disulfide peptide C49A-ss-C1B and intra-disulfide linked peptide C3B-ss-C5B were identified, the C2B situated between them was not observed probably due to the intrinsically low MS response. 

### 2.5. Novel Peptide Markers of Ricin from Other Proteinase Digestion

Peptide markers produced from other proteinase digestions are essential for verification of trace ricin in complex matrices. Here, two other proteases of pepsin and proteinase K were also attempted to digest ricin into potential peptide markers. 

#### 2.5.1. Novel Peptide Markers of Ricin from Pepsin Digestion

Pepsin is a protease with high digestion efficiency, which preferentially cleaves phenylalanine (F), leucine (L), and tyrosine (Y) at the C-terminus [21]. According to the product instruction, the specificity for cleavage at F and L is best at pH 1.3. In case the pH is above 2.0, the specificity for F and L cleavage decreases heavily with the increased specificity for Y. When the pH is above 5.0, the pepsin will lose its activity. Pepsin can be used alone or in combination with other proteases for protein analysis by mass spectrometry or other applications [21]. Here, ricin was separately digested with pepsin under the two conditions of pH 1~2 (0.67% FA) and pH 2~4 (0.1% FA) and then evaluated. Following theoretical enzyme digestion and specificity assessment by NCBI BLASTp search, the specific pepsin digest peptides were obtained in pH 1~2 (Appendix A) and pH 2~4 (Appendix A). During the digestion of ricin at pH 1~2, ten peptides from A chain and nine peptides from B chain were unique to ricin. At pH 2~4, there are only seven unique peptides and all of them are from the B chain. This phenomenon should be resulted from the reduced cleavage selectivity of pepsin at pH value above 2. Under the pH value, ricin preferred being cleaved into short peptides because of pepsin’s more cleavage sites of F, L, Y, E, Q, and A. 

Under the optimized conditions at pH 1~2 according to the procedure of 3.2.6, 7 out of 10 ricin A-chain specific peptides were verified by LC-Q-TOF/MS (Figure 5). Three peptides, P24A, P13A, and P7A exhibits the best MS abundance and their corresponding LODs were 10 ng/mL, 20 ng/mL, and 100 ng/mL respectively, thus they can be considered as novel peptide markers. The AB chain disulfide peptide P36A-ss-P1B was not observed, probably because of its large molecular weight and disulfide bond. In the B-chain, all ricin specific peptides were identified except P7B that is a long peptide with 42 amino acid residues. We selected the three peptides of P26B, P23B and P6B as novel peptide markers based on their highest MS intensity among specific peptides with the corresponding LODs of 10 ng/mL, 20 ng/mL, and 50 ng/mL. At pH 2~4, all seven ricin unique peptides were identified including disulfide peptide P2B-ss-P4B as showed in Figure 6 and listed in the Appendix A. These identified unique peptides form B-chain at pH 2~4 will not be enough for unambiguous identification of ricin with both chains. As a consequence, it is recommended to digest ricin with pepsin at pH 1~2 for its unique identification and quantification.

#### 2.5.2. Novel Peptide Markers of Ricin from Proteinase K Digestion

We also used proteinase K isolated from *Tritirachium album* to digest ricin into novel peptide markers. Under the optimized conditions in 3.2.7, one A-chain and five B-chain peptides were confirmed as ricin-specific peptides by in silico digestion and a BLASTp search (Table 3). All these peptides were identified by MS with good intensity (Appendix A). The LOD of ricin for K71A, K15B, K41B, K61B and K92B were 100 ng/mL, 50 ng/mL, 10 ng/mL, 50 ng/mL and 10 ng/mL, respectively. In spite of the limited number of peptide markers, they may have good sensitivity and cost short digestion time. Thus, these peptide markers can be used as a supplement for unambiguous identification of ricin.

## 3. Materials and Methods 

### 3.1. Sequence Alignment and Identification of Uniqueness of Digested Peptides

The amino acid sequence of ricin was downloaded from NCBI UniProt database (P02879). In silico digestion of ricin into particular peptides by various proteinases were generated using ExPASy peptide mass tool (https://web.expasy.org/peptide mass/). The in silico enzymatic peptides were selected under the following criteria: no missed cleavages, more than five amino acid numbers. The uniqueness of the digested peptides were identified by BLASTp searching them against the UniProtKB/SwissProt protein sequence database (~470,034 protein sequences). 

### 3.2. Digestion of Ricin Using Multiple Proteinases

#### 3.2.1. Trypsin Digestion of Ricin with Denaturation and Reduction

Fifty microliter of 0.2 mg/mL purified ricin was transferred into a 200 μL PCR tube, containing 42 μL of the denaturing and reducing solution composed of 20 μL aliquot of acetonitrile, 1 μL of 0.5 M dithiothreitol (DTT), 5 μL of 1.0 M ammonium bicarbonate (pH 8.5) and 16 μL of H_2_O. The mixture was incubated at 60 °C for 50 min and then cooled at room temperature for 10 min. 4 μg of trypsin/lys-C mix (Promega, Mass Spec Grade, Madison, WI, USA) was added and then the reaction solution was kept at 40 °C for 4 h. The reaction was terminated by adding the appropriate volume of 10% formic acid (FA) to achieve 0.5% final concentration and transferred to vials for LC-MS analysis.

#### 3.2.2. Solvent-Assisted Direct Digestion of Ricin

Digestion was performed according to our unpublished acetonitrile-assisted trypsin digestion method. Briefly, 50 μL of 0.2 mg/mL of ricin were transferred to a 200 μL PCR Tube, which contains 10 μL of acetonitrile and 27 μL of H_2_O. Then, 5 μL of 1.0 M ammonium bicarbonate (pH 8.8) and 4 μg of trypsin/lys-C mix (Promega, Mass Spec Grade) were added and incubated at 40 °C for 4 h. After digestion, the reaction was terminated by adding an appropriate volume of 10% FA to 0.5% final concentration. The resulting samples were transferred to vials for LC-MS analysis.

#### 3.2.3. Tandem Digestion with Glu-C after Trypsin

After solvent-assisted trypsin digestion of ricin according to the procedure of Section 3.2.2, the digestion mixture was concentrated to dryness using vacuum concentrator. Then, the sample was reconstituted with 80 μL 20 mM PBS (15 mM NaH_2_PO_4_ and 5 mM Na_2_HPO_4_). A total of 8 μL of 0.25 μg/μL aqueous Glu-C (Promega, Mass Spec Grade) solution was added, and the digestion was performed at 45 °C for 4 h. Five μL of 10% FA was added to terminate the digestion and then transferred to vials for LC-MS analysis.

#### 3.2.4. Chymotrypsin Digestion with Denaturation and Reduction

Fifty μL of 0.2 mg/mL of ricin were added to a 200 μL PCR tube, containing the denaturing and reducing solution composed of 20 μL aliquot of acetonitrile, 1 μL of 0.5 M dithiothreitol (DTT), and 10 μL of 1.0 M tris-HCl (pH 8.0), and 11 μL of H_2_O. The mixture was then incubated at 60 °C for 50 min. Following the sample being cooled to room temperature, 8 μL of 0.25 μg/μL chymotrypsin (Promega, Sequence Grade) was added and incubated at 40 °C for 4 h. After digestion, samples were acidified by adding 8 μL of 5% FA and transferred to vials for LC-MS analysis.

#### 3.2.5. Solvent-Assisted Chymotrypsin Digestion of Ricin

Firstly, 50 μL of purified ricin sample was transferred in a 200 μL PCR tube, which contains 10 μL of 1.0 M Tris-HCl (pH 8.0) and 10 μL acetonitrile. Subsequently, aliquot of 8 μL of 0.25 μg/μL chymotrypsin (Promega, Sequencing Grade) was added and then the mixture was incubated for 4 h at 40 °C. The digestion was terminated by adding 8 μL of 5% FA. The resulting sample was transferred to an autosampler vial for LC-MS analysis.

#### 3.2.6. Pepsin Digestion

Briefly, 12 μL of 2 mg/mL pepsin (Sigma-Aldrich) was added to 50 μL of 0.2 mg/mL of ricin in H_2_O. As for the digestion reaction under pH 1~2, 7 μL of 10% FA was added to the sample. As for that under pH 2~4, 1 μL of 10% FA was added. At last, the mixture was diluted to 100 μL with H_2_O. The mixture was incubated at 60 °C for 4 h. After digestion, the mixture reaction was terminated by adding an appropriate volume of 2% ammonium hydroxide and transferred to vials for LC-MS analysis.

#### 3.2.7. Proteinase K Digestion

Firstly, 20 μL of 20 mg/mL proteinase K (Merck Millipore, Darmstadt, Germany) in 20 mM Tris-HCl (pH 7.8) were added to 50 μL of 0.2 mg/mL of ricin in H_2_O. Then, 2 μL of 1.0 M Tris-HCl and 28 μL of H_2_O were added. The sample was mixed and incubated for 2 h at 60 °C. After digestion, 10% FA was added in the sample to terminate the reaction and the reaction solution was transferred to vials for LC-MS analysis.

### 3.3. Accurate Mass LC-MS Qualitative Analysis

An Agilent 1200 HPLC system combined with a 6520 high-resolution accurate Q-TOF mass spectrometer (Agilent technology, Palo Alto, MA, USA) was used for sample analysis. The digests peptides were verified by extract the accurate molecular weight of the in silico digested peptides. Sample aliquots were separated on an Advanced Peptide Map 2.7 μm, 2.1 × 150 mm column (Agilent technology, MA, USA). Mobile phase A consisted of 0.1% formic acid in water. Mobile phase B consisted of 0.1% formic acid in acetonitrile. The flow rate for analysis was 250 μL/min and the injection volume was 5 μL. The optimized gradient elution program is as follows: 0–3 min 5% B; 3–45 min linear gradient to 70% B; 45–53 min 70% B; 53–54 min linear gradient to 100%B; 54–60 min, 100%B. The column was re-equilibrated with 5%B for 4 min before the next run. Eluted peptides were injected to the mass spectrometer equipped with electrospray ionization source in positive mode MS scan was acquired over the mass range 150–3000 at 1.1 spectra/s; gas temperature = 330 °C, gas flow = 8 L/min, nebulizer = 30 psi, and capillary voltage = 3500 V; Fragmentor is 150 V and skimmer is 65 V. Mass Hunter workstation B.05.00 software was used (Agilent Scientific, USA) for instrument control and to process the data files.

## 4. Conclusions

In this work, tandem and multiple protease digestion method were developed for screening and identification of novel peptide markers of ricin using LC high-resolution MS analysis. In silico BLASTp database search was combined with accurate mass LC-MS analysis to identify and select novel marker peptides. We obtained more novel peptide markers than the previous existing peptide markers as shown in Table 4 and Appendix A. This study provides a new strategy for unambiguous identification of ricin during the investigation of illegal use of the toxin. 

## Figures and Tables

**Figure 1 toxins-11-00393-f001:**
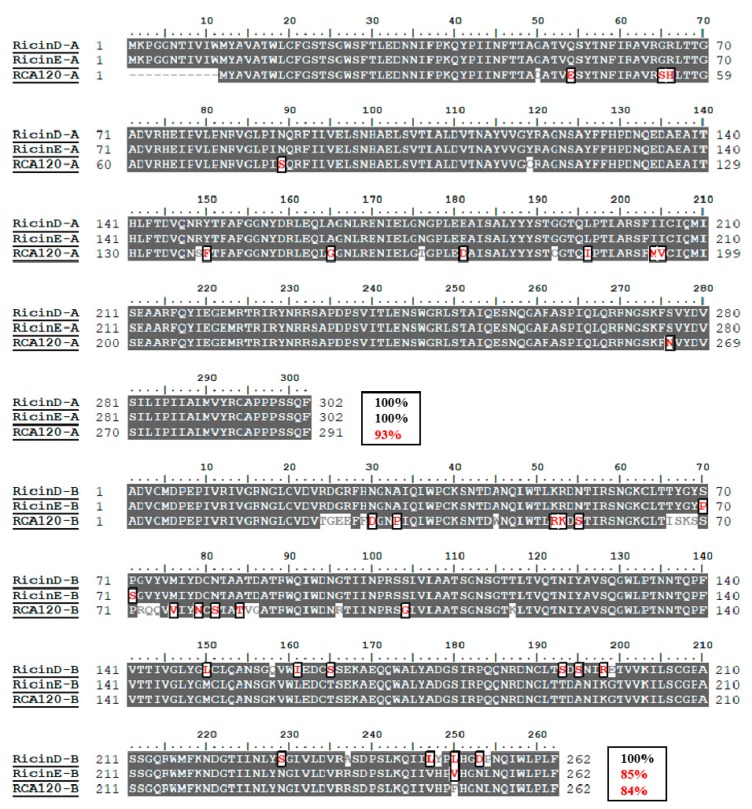
The amino sequence alignment of ricin D (Uniprot ID: P02879), ricin E (B chain GI:225419) and RCA120 (Uniprot ID: P06750) using the Clustal W program of BioEdit 7.0.9.0. Gray shading represents 40% identical residues among the sequences. Amino acids that differ among ricin D, ricin E and RCA120 are indicated by the outline. The amino acid similarity was also outlined at the end of each chain.

**Figure 2 toxins-11-00393-f002:**
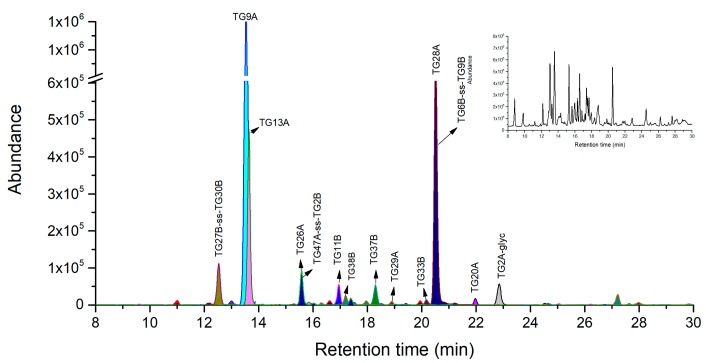
LC-high-resolution accurate mass analysis of the trypsin combined with Glu-C digest of 0.1 mg/mL of purified ricin base peak chromatogram overlaid on extracted ion chromatograms of ricin peptide makers. Total ion chromatography of ricin in the top right corner window, the original pictures were presented in Appendix A.

**Figure 3 toxins-11-00393-f003:**
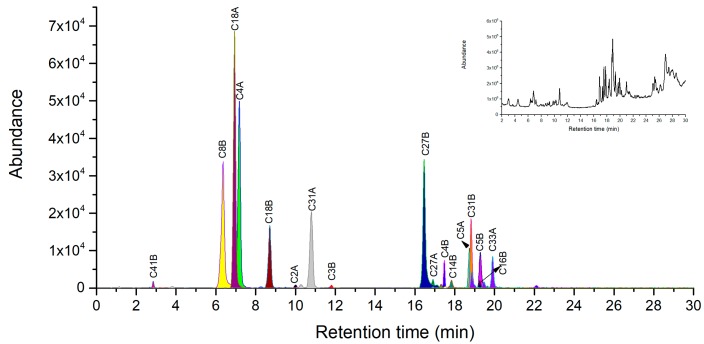
LC-high resolution accurate mass analysis of the chymotrypsin digest of 0.1 mg/mL of purified ricin under denaturation and reduction. Base peak chromatogram overlaid on extracted ion chromatograms of ricin maker peptides. Total ion chromatography of ricin at top right corner window, the original pictures were presented in Appendix A.

**Figure 4 toxins-11-00393-f004:**
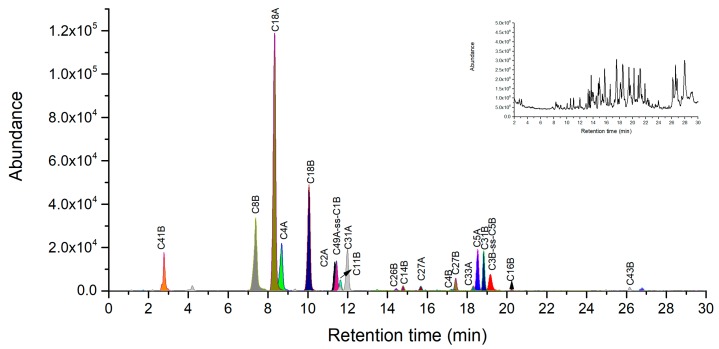
LC-high resolution accurate mass analysis of the direct chymotrypsin digestion of 0.1 mg/mL of ricin. Base peak chromatogram overlaid on extracted ion chromatograms of ricin maker peptides. Total ion chromatography of ricin at top right corner window, the original pictures were presented in Appendix A.

**Figure 5 toxins-11-00393-f005:**
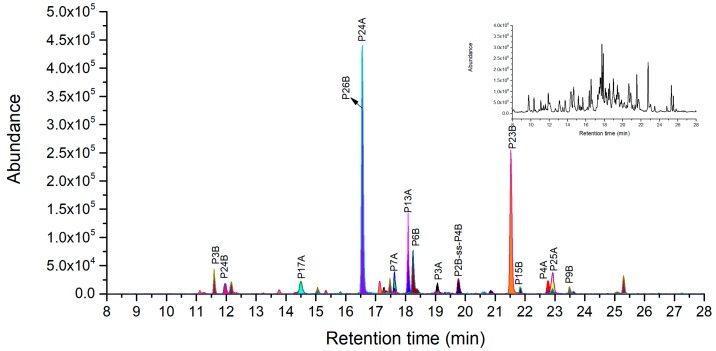
LC-high resolution accurate mass analysis of the pepsin digestion production of 0.1 mg/mL of purified ricin (pH 1~2). Base peak chromatogram overlaid on extracted ion chromatograms of ricin maker peptides. Total ion chromatography of ricin at the top right corner window, the original pictures were presented in Appendix A.

**Figure 6 toxins-11-00393-f006:**
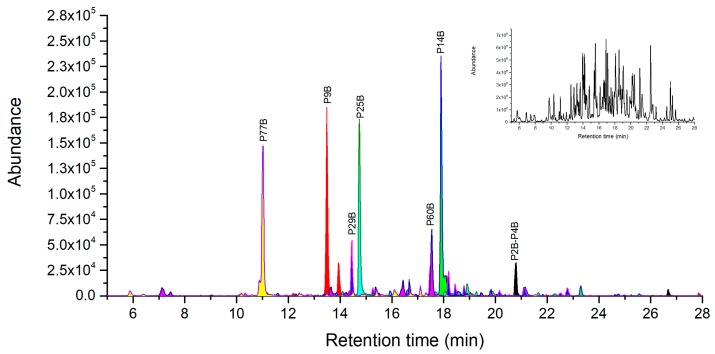
LC-high resolution accurate mass analysis of the pepsin digestion of 0.1 mg/mL of purified ricin (pH 2~4). Base peak chromatogram overlaid on extracted ion chromatograms of ricin maker peptides. Total ion chromatography of ricin at the top right corner window, the original pictures were presented in Appendix A.

**Table 1 toxins-11-00393-t001:** The specific ricin peptide markers under direct digestion with Glu-C after trypsin.

TG#&Chain	Amino Acid Sequence	(M + H)^+^	(M + 2H)^2+^	(M + 3H)^3+^	(M + 4H)^4+^
TG2A-glyc *	QYPIINFTTAGATVQSYTNFIR	3675.6946	1838.3509	**1225.9031**	919.6791
TG9A	VGLPINQR	896.5312	**448.7692**	299.5152	-
TG13A	VTNAYVVGYR	1141.6000	**571.3036**	381.2048	**-**
TG20A	YTFAFGGNYD	1154.4789	**577.7431**	385.4978	**-**
TG26A	LGNGPLE	**699.3671**	350.1872	233.7939	**-**
TG28A	AISALYYYSTGGTQLPTLAR	2146.1179	1073.5626	**716.0441**	537.2849
TG29A	SFIICIQMISE	1283.6374	**642.3223**	428.5506	**-**
TG47A-ss-TG2B ^a^	CAPPPSSQF-ss-VCMD	1397.5535	**699.2804**	466.5226	**-**
TG6B-ss-TG9B ^a^	NGLCVD-ss-FHNGNAIQLWPCK	2145.0003	1073.0038	**715.7154**	537.0555
TG11B	ANQLWTLK	973.5465	**487.2769**	325.1870	**-**
TG15B-ss-TG16B ^a^	CLTTYGYSPGVYVMIYD-ss-CNTAATD	2637.1194	1319.0633	879.7113	660.0353
TG20B	SSLVLAATSGNSGTTLTVQTNIYAVSQGWLPTNNTQPFVTTIVGLYGLCLQANSGQVWIED	6284.1859	3142.5966	2095.4002	1571.8024
TG27B-ss-TG30B ^a^	NCLTSD-ss-ILSCGPASSGQR	1824.7458	912.9094	608.9639	**456.9912**
TG33B	GTILNLYSGLVLD	1377.7624	689.3848	**459.9256**	**-**
TG37B	QIILYPLHGD	1168.6360	**584.8217**	390.2169	**-**
TG38B	PNQIWLPLF	1127.6248	**564.3160**	376.5464	**-**

# Trypsin/Glu-C tandem digest peptides numbered from the amino terminal of the polypeptide chain. * Main glycopeptide. ^a^ Disulfide bound peptide. The LC-MS observed ions were indicated in bold.

**Table 2 toxins-11-00393-t002:** The ricin specific chymotryptic peptides under digestion with denaturation and reduction.

C# and Chain	Amino Acid Sequence	(M + H)^+^	(M + 2H)^2+^	(M + 3H)^3+^	(M + 4H)^4+^
C2A	TTAGATVQSY	998.4789	**499.7431**	333.4978	-
C4A	IRAVRGRL	940.6163	470.8118	**314.2103**	-
C5A	TTGADVRHEIPVLPNRVGLP INQRF	2799.5376	1400.2724	933.8507	**700.6418**
C18A	TDVQNRY	895.4268	**448.2170**	299.1471	-
C27A	EEAISAL	**732.3774**	366.6923	265.1449	-
C31A	STGGTQLPTL	974.5153	**487.7613**	325.5099	-
C33A	IICIQM	**720.3783**	360.6928	240.7976	-
C2B	DPEPIVRIVGRNGL	1534.8699	767.9386	512.2948	384.4734
C3B	CVDVRDGRF	1066.5098	533.7585	**356.1748**	-
C4B	HNGNAIQL	**866.4479**	433.7276	289.4875	-
C5B	WPCKSNTDANQL	1376.6263	**688.8168**	459.5469	344.9125
C8B	KRDNTIRSNGKCL	1504.8012	752.9043	**502.2719**	376.9562
C14B	DCNTAATDATRW	**1324.5586**	643.8386	442.1911	331.8956
C16B	DNGTIINPRSSL	1286.6699	**643.8386**	429.5615	322.4234
C18B	AATSGNSGTTL	979.4691	**490.2382**	327.1612	-
C26B	QANSGQVW	**889.4162**	445.2118	297.1436	-
C27B	IEDCSSEKAEQQW	1552.6584	**776.8328**	518.2243	388.9205
C31B	TSDSNIRETVVKIL	1574.8748	787.9410	**525.6298**	394.4746
C40B	DVRASDPSL	959.4792	480.2432	320.4979	-
C41B	KQIIL	614.4235	**307.7154**	205.4794	-
C43B	HGDPNQIW	966.4428	483.7250	322.8191	-

# Chymotrypsin digest peptides numbered from the amino terminal of the polypeptide chain. The LC-MS observed ions were indicated in bold.

**Table 3 toxins-11-00393-t003:** The theoretical ricin peptides under proteinase K direct digestion.

K#&Chain	Amino Acid Sequence	(M + H)^+^	(M + 2H)^2+^	(M + 3H)^3+^	(M + 4H)^4+^
K71A	STGGTQL	663.3308	332.1690	**221.7818**	-
K15B	PCKSNTDA	835.3614	**418.1843**	279.1253	-
K41B	TSGNSGTTL	837.3948	419.2011	**279.8031**	-
K61B	EDCSSEKA	868.3353	**434.6713**	290.1166	-
K92B	HGDPNQI	780.3635	**390.6854**	260.7927	-

# Proteinase K digest peptides numbered from the amino terminal of the polypeptide chain. The LC-MS observed ions were indicated in bold.

**Table 4 toxins-11-00393-t004:** Peptide markers for ricin identification.

Peptide Marker	Amino Acid Sequence	(M + H)^+^	(M + 2H)^2+^	(M + 3H)^3+^	(M + 4H)^4+^
T7A	VGLPINQR	896.5312	**448.7692**	299.5152	-
T10A	YTFAFGGNYDR	1310.5800	**655.7936**	437.5315	-
T11A	LEQLAGNLR	1013.5738	**507.2905**	338.5294	-
T6B	SNTDANQLWTLK	1390.6961	**695.8517**	464.2369	-
T20B	QIILYPLHGDPNQIWLPLF	2277.2430	1139.1251	**759.7525**	570.0667
C2A	TTAGATVQSY	998.4789	**499.7431**	333.4978	-
C18A	TDVQNRY	895.4268	**448.2170**	299.1471	-
VL-C18B	VLAATSGNSGTTL				
TG13A	VTNAYVVGYR	1141.6000	**571.3036**	381.2048	**-**
TG26A	LGNGPLE	**699.3671**	350.1872	233.7939	**-**
TG28A	AISALYYYSTGGTQLPTLAR	2146.1179	1073.5626	**716.0441**	537.2849
TG6B-ss-TG9B ^a^	NGLCVD-ss-FHNGNAIQLWPCK	2145.0003	1073.0038	**715.7154**	537.0555
TG11B	ANQLWTLK	973.5465	**487.2769**	325.1870	**-**
TG27B-ss-TG30B ^a^	NCLTSD-ss-ILSCGPASSGQR	1824.7458	912.9094	608.9639	**456.9912**
TG37B	QIILYPLHGD	1168.6360	**584.8217**	390.2169	**-**
C4A	IRAVRGRL	940.6163	470.8118	**314.2103**	-
C31A	STGGTQLPTL	974.5153	**487.7613**	325.5099	-
C4B	HNGNAIQL	**866.4479**	433.7276	289.4875	-
C8B	KRDNTIRSNGKCL	1504.8012	752.9043	**502.2719**	376.9562
C18B	AATSGNSGTTL	979.4691	**490.2382**	327.1612	-
C27B	IEDCSSEKAEQQW	1552.6584	**776.8328**	518.2243	388.9205
C31B	TSDSNIRETVVKIL	1574.8748	787.9410	**525.6298**	394.4746
P7A	PINQRF	**774.4257**	387.7165	258.8134	-
P13A	DVTNAYVVGYRAGNSAYF	1966.9293	983.9683	**656.3146**	492.4883
P24A	EEAISAL	**732.3774**	366.6923	244.7973	-
P6B	KRDNTIRSNGKCL	1504.8012	752.9043	**502.2719**	376.9562
P23B	DVRASDPSL	959.4792	**480.2432**	320.4979	-
P26B	HGDPNQIWL	**1079.5268**	**540.2671**	360.5138	-
K71A	STGGTQL	663.3308	332.1690	**221.7818**	-
K15B	PCKSNTDA	835.3614	**418.1843**	279.1253	-
K41B	TSGNSGTTL	837.3948	419.2011	**279.8031**	-
K61B	EDCSSEKA	868.3353	**434.6713**	290.1166	-
K92B	HGDPNQI	780.3635	**390.6854**	260.7927	-

The existed marker peptides were indicated in black and the novel peptides are indicated in red.

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
