# Peer review of "LC-HRMS Screening and Identification of Novel Peptide Markers of Ricin Based on Multiple Protease Digestion Strategies"

_toxins, 2019, doi:10.3390/toxins11070393_

Round 1

Reviewer 1 Report

Excellent article!

I have no suggestions. 

Author Response

Point 1: Excellent article! I have no suggestions. Response 1: Thank you very much for your high evaluation of this manuscript.

Reviewer 2 Report

The authors describe strategy to obtain peptide markers for ricin identification. Most of the peptides predicted in silico are experimentally identified by LC-HRMS. The experiments are correctly conducted and most of the literature on ricin identification is cited.

- However the digestion is done only on purified ricin and it may be interesting to present results of more real cases such as identification from digestion of complex mixture as provided in EQuATox (reference 17)

- it should be interesting to give the sensibility of ricin detection for different peptide markers

- A conclusion is lacking at the end of the manuscript. It could be useful to provide a summary table comparing peptide markers already in use and new peptides presented in this work 

Minor points:

Some mispelling

- line 13 was used to veriified

- line 155: 713A was cleavage

 - line 192: and the intermolecular disulfide (it should be intramolecular)

Reviewer 3 Report

Ricin is a highly toxic protein composed of two chains, chain A and chain B, the functional N-glycosidase activity being carried by chain A. Ricin belongs to large family of proteins known as Ribosome-inactivating protein (RIP). Although they most probably share the same molecular mechanism on the ribosome, RIPs have variable toxicities. Indeed, ricin is much more toxic than R. communis agglutinin (RCA120) although they have high sequence homology rates. Therefore, it is critical to found reliable methods to unambiguously discriminate Ricin from RCA120. In this manuscript, the authors described a novel method to analyse Ricin by Mass Spectrometry. By performing distinct proteolytic disgestions such as Glu-C digestion after trypsin treatment, chymotrypsin digestion, chymotrypsin digestion under denaturation and reduction, pepsin and proteinase K digestions, the authors managed to isolate and analyse by Mass Spectrometry several discriminative peptides markers from Ricin. The authors performed many experiments on pure Ricin that are generally well designed. The overall obtained data are generally compelling and should allow accurate characterization of Ricin. To strengthen the manuscript, this reviewer suggests that the authors should validate their conclusions by performing at least one of the proposed digestions on RCA120 and compare it to Ricin in order to demonstrate that their experimental strategy would unambiguously allow the discrimination of RCA120 from Ricin. With this additional experiment, this manuscript is suitable for publication in Toxins. In addition, highlighting the Ricin peptide markers on the peptide sequence alignments of figure 1 would significantly improve the clarity of the manuscript.    

Minor point

In table S1, the length of T12B peptide is 42 aa, in the text, it is mentioned as 67 aa (p4, lane 114).

Reviewer 4 Report

The Title clearly identifies the article. The Abstract describes its content briefly. The Results and Discussion section (including Supplementary material) shows the all information in clear and understandable way. Therefore in my opinion the presented manuscript could be accepted with the minor revision as an Article to Toxins.

1.      Line 65: “proteases” rather than “proteinases”, please check the line 231 also

2.      Line 66: “positive MS” rather than “good MS”

3.      Line 69: …acid sequence between ricin...

4.      In all manuscript: Ricin…or ricin? and “Acn” rather then “ACN”; Proteinase K or proteinase K; please check the font of “ºC”

5.      Line 126-127 – space

6.      Line 220: m/z

7.      Figures 2-5: the total ion chromatographies of ricin are illegible.

8.      The References please prepare according to Instructions to Authors.

Round 2

Reviewer 3 Report

The authors carefully properly addressed all my concerns about their manuscript. Therefore, this reviewer recommends the revised version of this manuscript for publication in Toxins.